# Synergistic patient factors are driving recent increased pediatric urgent care demand

**Emily Lehan**[1], **Peyton Briand**[1,2], **Eileen O'Brien** [1,2], **Aleena Amjad Hafeez**[1], **Daniel J. Mulder** [1,2]*

1 Department of Pediatrics, Queen's University, Kingston, Ontario, Canada, 2 Department of Biomedical and Molecular Sciences, Department of Medicine, Gastrointestinal Diseases Research Unit, Queen's University, Kingston, Ontario, Canada

* daniel.mulder@queensu.ca

**Data Availability Statement:** Data cannot be shared publicly because it contains personal health information. Data are available from the Canadian

## Abstract

### Objectives

We aimed to use the high fidelity urgent care patient data to model the factors that have led to the increased demand at our local pediatric urgent care centre.

### Methods

The dataset for this retrospective cohort study was obtained from our local healthcare centre's national reporting data for pediatric urgent care visits from 2006 to 2022. Variables analyzed included: basic patient demographics, chief complaint, triage urgency, date and time of registration/discharge, discharge diagnosis, and discharge destination. Statistical analysis of non-linear trends was summarized by locally estimated scatterplot smoothing splines. For machine learning, we used the tidymodels R package. Models were validated in training using k-fold cross validation with k = 5. We used univariate linear regression as a baseline model. After the data was standardized, correlation and homoscedasticity were evaluated between all parameter permutations.

### Results

This dataset consisted of 164,660 unique visits to our academic centre's pediatric urgent care. Over the study period, there was an overall substantial increase in the number of urgent care visits per day, with a rapid increase beyond previous levels in 2021 and further in 2022. The increased length of stay trend was consistent across presenting complaint categories. The proportion of patients without primary care in 2022 was 2.5 times higher than in 2013. A random forest machine learning model revealed the relative importance of features to predicting a visit in 2022 were: longer stay, later registration in the day, diagnosis of an infectious illness, and younger age.

### Conclusions

This study identified a combination of declining primary care access, circulating viral infections, and shifting chief complaints as factors driving the recent increase in frequency and duration of visits to our urgent care service.

Institutes for Health Information (CIHI) for researchers who meet the criteria for access to confidential data. Applications to access the raw data used for this project can be made via the Canadian Institute for Health Information (cihi.ca, help@cihi.ca).

**Funding:** The author(s) received no specific funding for this work.

**Competing interests:** The authors have declared that no competing interests exist.

## Author summary

This study leverages a large dataset of pediatric urgent care visits to demonstrate the complex interplay of factors driving the recent increased care burden. The patterns in the data are not always readily evident with standard statistical techniques, but using machine learning helps discern differences not previously evident. A combination of declining primary care access, circulating viral infections, and more urgent complex presentations appear to be driving the recent increase in frequency and duration of visits to our urgent care service. These increases began prior to the COVID-19 pandemic in 2020, but appear to have been exacerbated by this event.

## Introduction

Emergency department (ED) volumes have recently increased across the developed world, including for children [1–3]. Increased ED wait times and overcrowding are known to increase mortality and threaten patient safety [4–5]. Although the issue of increased emergency department use is multifaceted, two important recent shifts in patient characteristics are notable: lack of access to primary care services and the COVID-19 pandemic [2,6]. The rising burden of care may be secondary not just to increased absolute number of emergency visits, but also from increased length of stay [7]. It is unclear what demographic and/or patient-related factors are driving these changes. How these factors have altered the pediatric emergency care demands is poorly understood.

The Canadian healthcare system is decentralized and publicly funded, with province-level administration. Canadian healthcare facilities are thus required to provide standard highly structured data reports for day surgery, outpatient clinics, and emergency departments through the National Ambulatory Care Reporting System (NACRS) [8]. This data is relatively standardized and of high quality, allowing for evaluation of patient and system factors that may be leading to the recent shifts in patient care challenges for children. Improved understanding of the underlying factors driving recent patient care challenges could improve resource allocation and direct public health interventions.

Patterns in routinely collected clinical data are difficult to uncover. Increasing availability of large multiparameter clinical databases has driven the adoption of machine learning (ML) techniques for analysis of multiparameter clinical data [9]. One powerful technique to identify patterns that are not obvious when interpreted by a human is machine learning (ML). ML is the application of algorithms to data analysis in such a way that the accuracy of the analysis can be improved by automated iteration [10]. The computational power of modern computers facilitates the use of ML algorithms to discern patterns in clinical data that are complex and non-linear [10].

In this study, we retrospectively analyzed a large longitudinal dataset of pediatric urgent care visits to determine the the factors that have led to the increased pediatric urgent care demand. Time series analysis and machine learning was used to uncover patient factors that changed as demand increased. We hypothesized that patient factors have shifted over time driving the recent increased pediatric urgent care service demand.

## Methods

The dataset for this retrospective cohort study was obtained from our local healthcare centre's NACRS reporting data. The data was collected from the electronic health record database

(EHR) for our hospital centre that is used for mandatory reporting to the Ministry of Health. This data is reviewed by our Decision Support teams prior to reporting, so it is of relatively high quality. Thus, we extracted only data fields from the EHR that are cleaned and reported to the Ministry. Applications to access the raw data used for this project can be made via the Canadian Institute for Health Information (cihi.ca, help@cihi.ca). The urgent care is a walk-in service open weekdays that cares for any patient under 18 years of age, regardless of healthcare coverage, where they can be seen by a pediatrician without an appointment. There are no other pediatric specialized urgent care or emergency department services in the region, making trends found in this dataset highly reflective of local healthcare needs. When the urgent care centre is closed, pediatric patients are seen in the local emergency departments (that are not pediatric specialized).

The study cohort consisted of all consecutive patient encounters obtained from our local healthcare centre's pediatric urgent care NACRS retrospective monitoring data between April 1, 2006 to Dec 31, 2022. The inclusion criteria were patients under 18 years of age, seen at the urgent care centre within the study timeline, regardless of healthcare coverage status. Exclusion criteria were any of: any incomplete demographics or clinical details in the extracted fields listed above or the visit was begun more than 60 min before the scheduled opening time (28 visits), the patient was discharged prior to the scheduled opening time (13 visits), the patient left more than 15 hours after the scheduled opening time (which is midnight, 124 visits), and if the length of stay was greater than 1000 min (that is 16 hours and 40 minutes; 13,651 visits). Given that our urgent care centre is only staffed for up to about 12 hours (and then is closed overnight), records with very long visit times represented patients who were missed as being recorded discharged in the health record and then coded as discharged at the start of the next day, which is a known clerical work around used in our system.

Variables analyzed included: basic patient demographics (age, sex, primary care status yes/no/unknown), chief complaint, triage urgency (using the standardized Canadian Triage and Acuity Scale, CTAS), date and time of registration, time of discharge, discharge diagnosis, and discharge destination (home/admission/left without being seen). Records were removed if there was any incomplete demographics or clinical details in the extracted fields. Records were filtered out if: the visit was begun more than 60 min before the scheduled opening time (28 visits), the patient was discharged prior to the scheduled opening time (13 visits), the patient left more than 15 hours after the scheduled opening time (which is midnight, 124 visits), and if the length of stay was greater than 1000 min (13,651 visits). From the filtered dataset, we calculated for each visit: the day of the week, season, and the length of stay.

Our objective was to use high-fidelity NACRS data to model and identify factors contributing to the increased demand for pediatric urgent care at our local urgent care center. To investigate this, we employed an observational retrospective study design, analyzing data collected through NACRS. We analyzed the demographic, clinical, and healthcare system characteristics of patients attending the pediatric urgent care centre. Statistical methods included bivariate and multivariate analyses to identify factors contributing to the increased demand.

Data analysis was performed using the R programming language (version 4.3.1). Statistical analysis of non-linear trends was summarized by locally estimated scatterplot smoothing (LOESS) splines using the *ggplot2* package (v3.4). For machine learning, we used the tidymodels package (v1.1) using the developers' recommended approach [11]. Each dataset was divided 80:20 into training and testing subsets. Testing datasets were only evaluated once, after the conclusion of the development of the model on the training set. Models were validated in training using k-fold cross validation with k = 5. We used univariate linear regression as a baseline model. Hyperparameter tuning used a hybrid approach of a combination of broad grid search and then focused iterative search.

The length of stay variable, when used as an outcome, was log-10 transformed. This parameter was highly right skewed and thus transformation decreased the impact of outliers on the model (which were often complex patients with unique reasons for prolonged length of stay, S1 Fig). After the data was standardized, correlation and homoscedasticity were evaluated between all parameter permutations. Full details of the statistical and modelling packages used are available in S1 File. Analysis scripts are available at https://github.com/DanJMulder/urgentcare.

### Ethics statement

This study was approved by our institution's health science research ethics board (PAED-546-22). This study only included anonymized retrospective patient data and thus informed consent was not possible to obtain.

## Results

A total of 164,660 visits were included in the analysis. The median visit length was 85 minutes with a range of 1 to 667 minutes. Age was consistent across the study periods with a linear regression p-value of 0.14 for age (in years) compared to year of visit. Sex distribution was similarly consistent over the study period with a Pearson's chi-squared p-value of 0.06. The relative disposition of patients between discharge home, admission, and "left without being seen" shifted significantly, with a Pearson's chi-squared p-value of $<2.2*10^{-16}$ due to an increased proportion of admissions since 2020, with a commensurate decrease in discharges home. The days per year our urgent care service was also consistent, with a linear regression model of days open per year p-value of 0.84. Demographic distribution over time during the relevant time periods of this study are summarized in Table 1.

Over the study period, there was an overall substantial increase in the number of visits per day, with daily volumes increasing in 2015 and then dropping again during the COVID-19 pandemic (in 2020) with a rapid increase beyond previous levels in 2021 and further in 2022 (Fig 1A). The average daily length of stay was stable from 2006 to 2021, despite increased visits per day, then in 2021 and further in 2022, the length of stay increased beyond previous years (Fig 1B). The total hours of care per day trend (calculated by totaling the length of stay for each visit for the day) increased to a new baseline in 2015, corresponding with the increased visit volume, and then trended up rapidly over 2021 and 2022 to daily levels consistently above previous maxima, especially during peak volume seasons (Fig 1C). When the same total hours of care data were totaled by calendar year, an increase in 2022 was evident (Fig 1D). The total hours of care in 2022 was 1.8 times the mean value of the all the previous years where there is complete data (2007–2021).

The distribution of length of stay in our pediatric urgent care was consistently less than 150 min, with a mean of 95.5 minutes and a skewness of 2.24 in 2014. The length of stay was

**Table 1. Summary of patient factors across the timeline of the study.**

|  | Visits 2007–2019 | Visits 2020–2021 | Visits 2022 |
|---|---|---|---|
| **Age in years, mean (SD)** | 5.75 (5.16) | 6.03 (5.38) | 5.70 (5.05) |
| **Sex, proportion female** | 0.48 | 0.49 | 0.48 |
| **Acuity (CTAS), mean (SD)** | 3.88 (0.61) | 3.80 (0.63) | 3.73 (0.62) |
| **Lack of primary care access, proportion** | 0.2 | 0.18 | 0.13 |
| **Length of stay in minutes, mean (SD)** | 95.4 (63.0) | 124.3 (75.6) | 155.8 (95.6) |

CTAS, Canadian Triage and Acuity Scale; SD, standard deviation

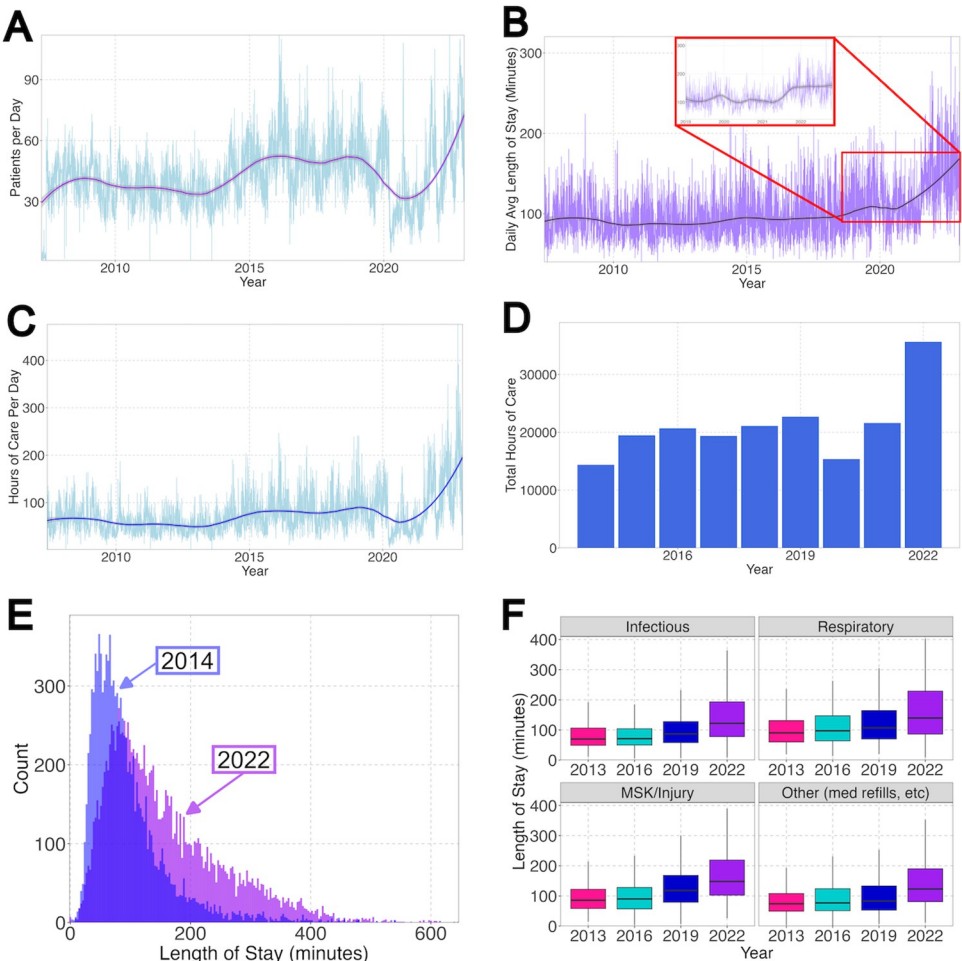

**Fig 1. A substantial increase in local pediatric urgent care demand.** The number of visits (A) and length of visits (B) increased over the study period, especially since 2021, with a new elevated baseline visit length (inset showing 2019–2022 trend). (C) The total hours of care per day nearly doubled from pre-2020 levels and the trend suggests the increase will continue into 2023. (D) Annual total of hours of care by calendar years show the increase for 2022 nearly doubled compared to that of any previous 8 years. (E) Visit length in 2014 was consistently less than 150 min with some outlier longer visits. The length of stay was consistently longer by 2022, with a larger spread of visit length, skewed toward longer visits. (F) Visit length increased consistently across the four most common diagnosis categories over the study period: infectious, respiratory, musculoskeletal, and other (which includes non-urgent visits such as those for medication refills and imaging follow up). *MSK, musculoskeletal.*

consistently longer by 2022, with a mean of 156.0 minutes and a right skewness of 1.11 (Fig 1E). The increased length of stay trend was consistent across the four most common diagnosis categories (infectious, respiratory, musculoskeletal, and other) (Fig 1F).

The distribution of discharge diagnoses shifted over the study period. Steadily less and less patients received "head and neck" category diagnoses (which include ear/nose/throat diagnoses and also dental and eye problems) and steadily more received infectious diagnoses (Fig 2A). Notably, the trend of an increased proportion of infectious presentations was first notable in 2018, *prior* to the COVID-19 pandemic. Urgency of triage acuity level increased over time, in addition to an absolute increase in CTAS 1 and 2 presentations ("resuscitation" and "emergent" level, respectively), the greatest changes over time were an increase in the proportion of CTAS 3 presentations ("urgent") (Fig 2B).

Decreasing primary care access paralleled the increase in visit number. The number of visits where the patient did not have an identified primary care practitioner (either designated at triage as "none" or "unknown"), increased steadily from 2014 to 2019 then decreased in 2020 and 2021 during the COVID-19 pandemic, then increased in 2022 to 1.4 times from the previous maximum level in 2019 (Fig 2C). Given the decrease in visits through the pandemic, it was important to assess beyond absolute numbers. Thus, we examined the proportion of patients presenting without primary care over the study period and found the proportion also steadily

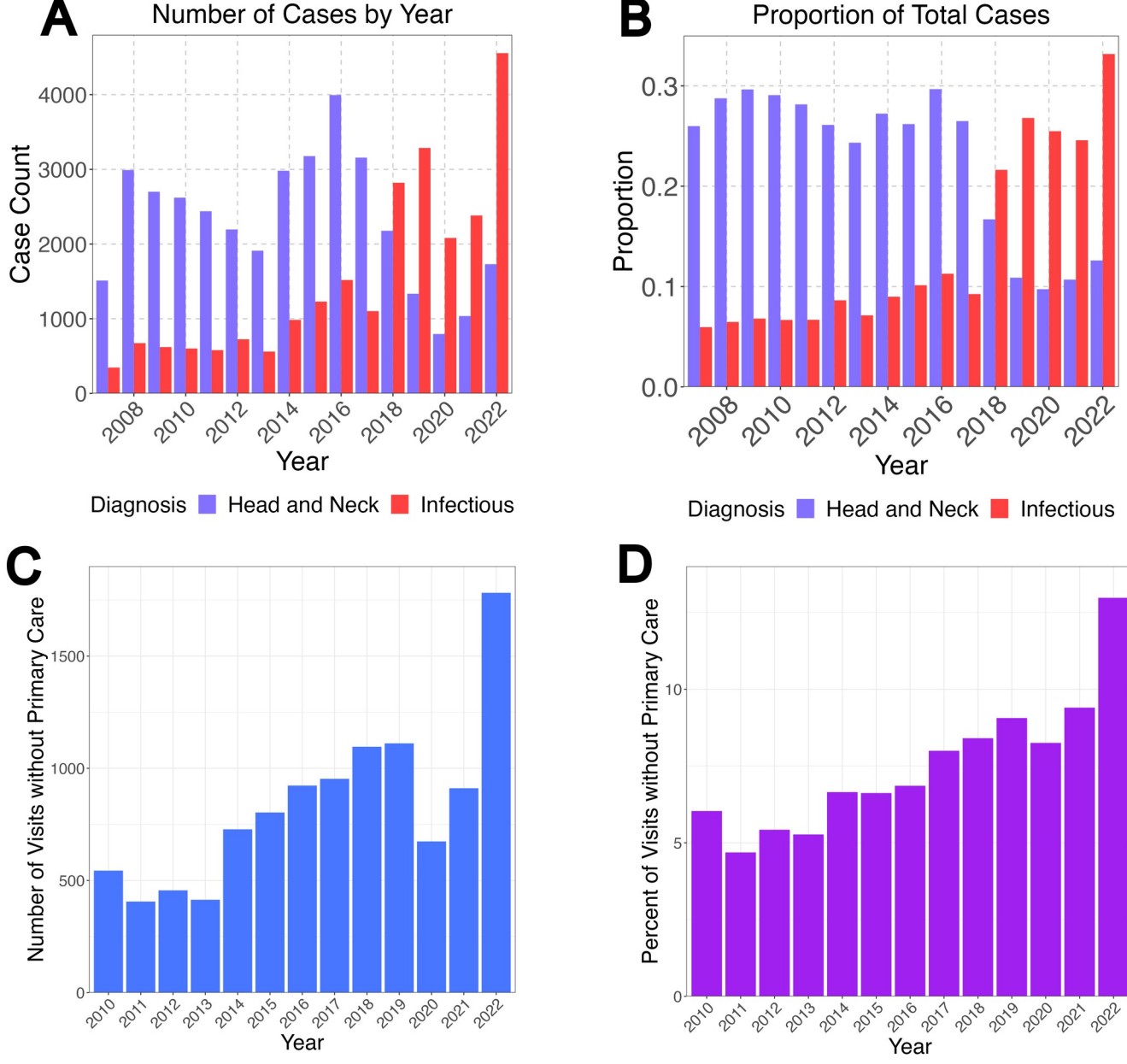

**Fig 2. Increased infectious diagnoses and urgent presentations over the study period.** Two notable contrasting changes in discharge diagnoses trends: head and neck diagnoses decreased in count and frequency (purple bars, A and B), while infectious diagnoses increased in count and frequency in a relatively commensurate amount (red bars, A and B). Increased urgent care service needs of those without primary care: the absolute number (C) and percent (D) of visits by those without primary care increased over the study period with 2022 having the highest demand for those without primary care access compared to all other years.

increased from 2014–2022 (with the exception of 2020) (Fig 2D). The increase in patients without primary care from 2021 to 2022 was from 9.1% to 13.0%. The proportion of patients presenting without primary care in 2022 was 2.5 times higher than in 2013.

There was an increase in the absolute numbers of all levels of acuity across the study period. The trend is most notable for "urgent" level presentations (CTAS 3), with more than triple the urgent presentations in 20022 when compared to 2007 (Fig 3A). The proportion of urgent presentations was also approximately steady between 2007 and 2018, ranging from 11% to 16% of the cases (Fig 3B). Then in 2019, the proportion of urgent presentations increased each year until 2022, when urgent presentations accounted for 24% of all cases.

Patients without identified primary care were more likely to present with "emergent" level and "non-urgent" level presentations (CTAS 2 and 5, respectively) (Fig 3C). Patients without primary care were also disproportionately more often diagnosed with mental health or non-medical diagnoses (Fig 3D). Notably, a relatively low proportion of patients without primary care left without being seen (assigned a diagnosis of "LWOBS"), likely indicating that even when volumes and wait times were high, these patients were more likely to persist waiting for non-urgent issues since their other avenues of access to basic medical care are limited. Individual comparison between the main study outcomes (visits in 2022 and length of stay) are summarized in Table 2.

For machine learning models, preprocessing consisted of removing all rows with empty values, normalization of all numeric features, and dummy variable creation for all nominal features. No numeric features were significantly correlated with any other. No features were significantly correlated with the outcome. The highest r-squared values for the length of stay were: year of visit (0.19) and time of discharge (0.42). Linear regression and random forest models were used to determine prediction of length of stay. The linear regression model was optimal, with a tendency to overestimate long visits and had an r-squared value of 0.838 with a root mean squared error (RMSE) of 0.108 (S2A–S2B Fig). The random forest model underestimated length of stay with an r-squared value of 0.830 and an RMSE of 0.133 (S2C–S2E Fig).

To determine the factors that contributed to 2022 having increased visits which were uniquely longer, we trained multiple machine learning algorithms on time series data with the aim of understanding the complex interplay of patient factors that were altered in 2022 compared to other years. A random forest model was optimal for predicting a 2022 visit with a receiver-operator curve area under the curve of 0.79 and an F1 score 0.96. F1 score was a preferred metric for these models since the dataset was severely imbalanced with an approximately 8:92 ratio in the outcome data (2022 to all other years). This was optimal when compared to other machine learning algorithm-based models including XGBoost, logistic regression, and k-nearest neighbours. While all five algorithms had high F1 values (Fig 4A), the random forest model had the highest receiver operator area under the curve (Fig 4B). A random forest machine learning model determined features more likely to be associated with a 2022 visit compared to other years. In decreasing level of feature contribution to the model were: a longer stay with a later visit in the day and a diagnosis of an infectious illness and younger age were features with high relative importance visits in 2022 (Fig 4C). Notably, the machine learning model identified primary care access as the seventh highest feature in relative importance to predicting a visit to be in 2022.

## Discussion

This study used over 17 years of data from this single centre pediatric urgent care to identify patient factors that have shifted over time driving the recent increased pediatric urgent care service needs. We found that declining access to primary care, increased circulating infectious

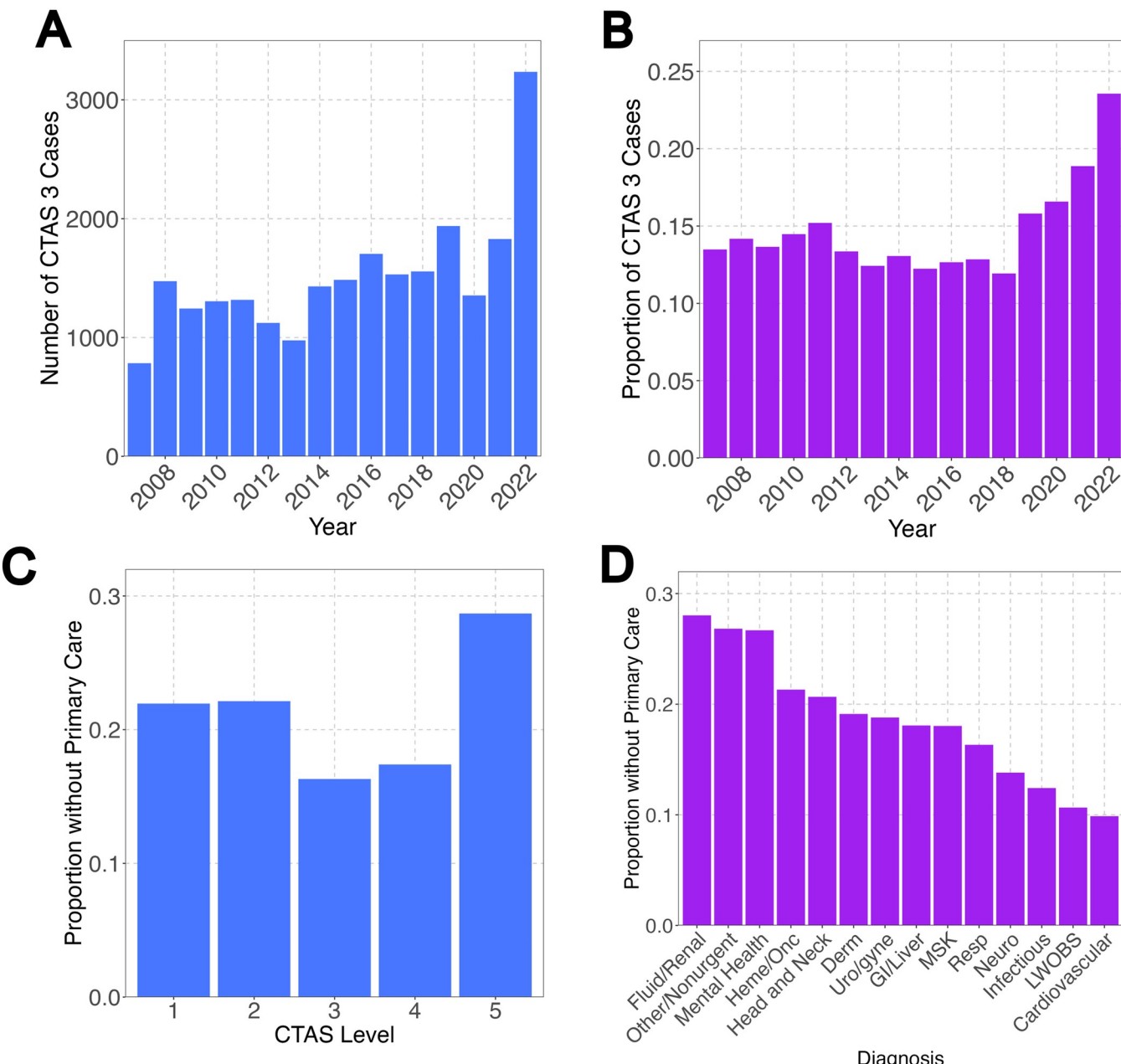

**Fig 3. Urgent and non-urgent patient presentations have increased especially for those without primary care, driven by specific diagnoses.** Increased number (A) and proportion (B) of "urgent" presentations over the study period as defined by the Canadian Emergency Department Triage and Acuity Scale level 3 (CTAS 3) assignment at triage. (C) Visits by acuity level show that patients without primary care are disproportionately more likely to present with triage assessment levels of "resuscitation" (CTAS 1) or "emergent" (CTAS 2) or "non-urgent" (CTAS 5). A lower proportion of those without primary care presented with triage levels of "urgent" (CTAS 4) or "semi-urgent" (CTAS 5). (D) Patients that did not have primary care access were disproportionately diagnosed with diagnosis categories of "fluid/renal", "other/non-urgent", and "mental health". *MSK, musculoskeletal; LWOBS, left without being seen. CTAS, Canadian Emergency Department Triage and Acuity Scale.*

diagnoses, and shifts in chief complaints are driving increased frequency and duration of visits. Machine learning models demonstrated the relative importance of patient factors responsible for a portion of these shifts.

Previous work provided a foundations for using statistical analysis combining trends over time and machine learning to predict length of stay for hospital inpatients [7,12]. A previously

**Table 2. Individual comparison of main outcome variables with patient factors.**

| Outcome Variable | Factor | Statistical Measure | Value[a] | p-value[b] |
|---|---|---|---|---|
| Visit in 2022 | Length of Stay | t-test | 68.4 | $<1*10^{-5}$ |
| | Age | t-test | -1.82 | $6.9*10^{-2}$ |
| | Sex | Chi-square | 2.04 | $1.5*10^{-1}$ |
| | Acuity (CTAS) | Chi-square | 1097 | $<1*10^{-5}$ |
| | Primary care access | Chi-square | 294 | $<1*10^{-5}$ |
| Length of Stay | Age | Pearson Correlation | 0.638 | $<1*10^{-5}$ |
| | Sex | t-test | 3.30 | $\mathbf{9.8*10^{-4}}$ |
| | Acuity (CTAS) | ANOVA | 10822 | $<1*10^{-5}$ |
| | Primary care access | t-test | 4.71 | $<1*10^{-5}$ |

a, for t-test value is t-statistic, for Chi-square test value is Chi-squared statistic, for Correlation value is rho-squared; b, values in bold if p-value <0.05
ANOVA, analysis of variance; CTAS, Canadian Triage and Acuity Scale

identified limitation in this type of study was the inherent complexity of patient-based data, although high levels of structure have enabled high model fidelity [13–15]. Our data set was highly structured since the patient data was cleaned and structured for our national reporting service. This study also benefitted from the relatively simple patient factors that inform care in pediatric urgent care, where patients usually have a single complaint and limited past medical history. The findings help to identify current deficiencies in the health care system that could be targeted for change. Declining access to primary care means patients may wait longer to seek medical care and may thus be presenting in a sicker state. Identifying an increase in infectious presentations means that preventive measures such as public health campaigns could target these needs. Additionally, the increase in patients with more complex needs may mean that longer visits are a new reality for urgent care centres.

Using a machine learning model to predict more recent visits is one approach to identify patient factors that have changed from previous years. A similar approach has been previously used to predict emergency department visits [16], and hospital admissions and outpatient visits [17]. Since 2022 was the most recent year for which we had available high-quality data, and because the trends explored in Figs 1–3 were strikingly different in 2022, we used 2022 visits as

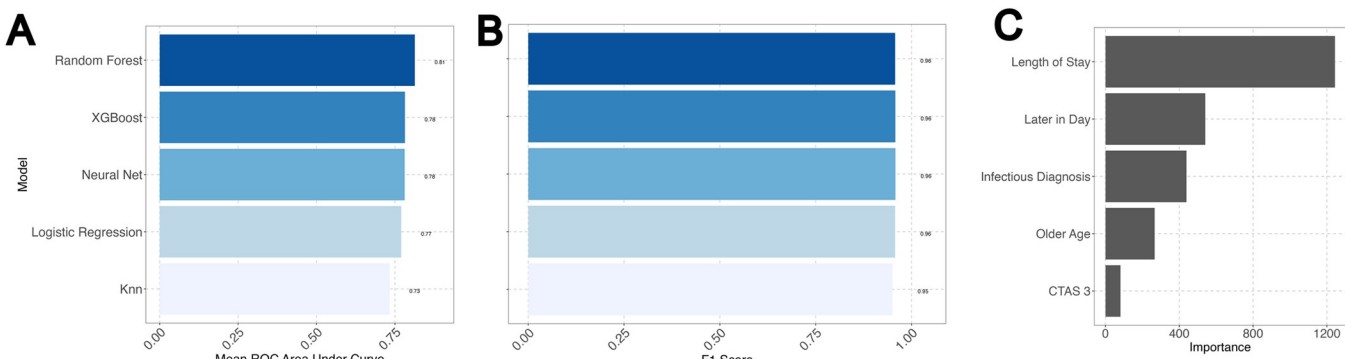

**Fig 4. A random forest model demonstrates the patient factors that were relatively different in 2022 compared to other years, identifying the factors that likely contributed to the increased service demand during 2022.** (A) Mean receiver-operator characteristic curve area-under-the-curve for testing set performance of five tuned machine learning models to assist understanding the complex interplay of patient factors that were altered in 2022 compared to other years. A random forest model was optimal for predicting a 2022 visit with a receiver-operator curve area under the curve of 0.79 (A) and an F1 score 0.96 (B). Features with the highest relative importance to predicting visits in 2022 were (in decreasing order): a longer stay, a visit later in the day, a diagnosis of an infectious illness, younger age, and a triage acuity of CTAS 3, "urgent" (C). *Knn, k-nearest neighbours.*

our outcome for this part of the analysis. We aimed to discover what patient factors were different in 2022 that could explain this difference. Thus, we built a model to identify important but subtle statistical differences between 2022 and previous years that would be unlikely to be evident from simple trend analysis.

Relatively simple statistical models were able to predict length of stay in this dataset with high fidelity, although the longer the visit, the less accurate the predictions were. This is similar to models developed from health record data to predict mortality and length of stay [17,18]. It should be noted, however, that previous studies either analyzed healthcare data trends prior to the COVID-19 pandemic or during it [16]. This study is a relatively new type of study that includes both data from prior to the COVID-19 pandemic and through it, up to the present. Although incorporating this broad timespan into statistical analysis likely adds complexity to the data, it reflects the current reality when attempting to predict trends and shifts in patient factors. Thus, our dataset provided a unique perspective on changing trends in pediatric urgent care needs, some shifts appeared to occur as a result of factors related to the pandemic. Importantly, some critical shifts, such as increasing infectious diagnoses and decreasing access to primary care, shifted prior to the pandemic but this shift has accelerated since, and appears likely to continue. These critical changes will be imperative to monitor in future studies.

Random forest models specifically are ideal for large organized digital health datasets such as the one analyzed in this study. Random forest models are: relatively efficient compared to more complex models such as neural networks, less prone to overfitting, and amenable to missing data (common concerns in human health data research). Thus, random forest models provide an excellent "out-of-the-box" ML model to use for clinical data analysis.

It is likely that the linear regression model tended to overestimate length for long visits as the visit length data was strongly left skewed, even when log-10 transformed (S1 Fig), thus the longer visits acted as outliners and were less accurately modelled. Linear regression models are known to be less accurate for outliers and skewed datasets. Despite this limitation, the large majority of visits were accurately predicted by the linear regression model (S2A Fig), but the longer less common visit lengths were overestimated (S2 Fig) by this model as the longer visits deviate from the normal. Thus, the longer the visit, the less accurate this relatively simple model can predict.

An important limitation of this study is that it is a single centre retrospective study so it could be susceptible to selection bias. This study was limited to a single centre in a medium size city serving a large regional population in Canada, which limits generalizability for some findings. However, the trends seen in patient characteristics and factors contributing to increased care needs would be readily applicable to many similar sized centres in North America. Another limitation is that this study relies on electronic health records, which are imperfect and can bias data in unexpected ways from incomplete or inaccurate recording. Certain aspects of the data may be influenced by regional shifts that are not as pronounced elsewhere. However, there is still merit in identifying the links between patient factors and increased service demands, which may occur in different ratios elsewhere, but the trends would still be likely generalizable to other populations. This limitation is counterbalanced by the strength of the highly internally consistent organized large dataset. Another limitation is that workload fluctuations, including nursing ratios and paperwork quantity, may have played a role in determining patient length of stay, although this data is not tracked at our centre. It would be helpful if these factors were added to the current reporting requirements, so these could be accurately evaluated in future studies.

In the present study, ML models were leveraged to analyze an extensive imperfect clinical dataset with numerous variable types. A traditional statistical approach would require simpler modelling that is less effective and less generalizable (as seen in the differences between models

in S2 Fig). The automation, efficiency, and scalability of ML models are strengths, especially for large data sets such as this.

This study identified a combination of declining primary care access, increasing circulating viral infections, and shifting chief complaints as factors driving the recent increase in frequency and duration of visits to our urgent care service. Understanding the evolving changes to patient factors that are influencing urgent care service needs can inform proactive resource allocation and improve outcomes.

## Supporting information

**S1 Fig.** Histogram of untransformed length of stay in minutes (A) compared to the more normalized (although still right skewed) distribution after log-10 transformation (B).
(DOCX)

**S2 Fig. Comparison of machine learning models to predict length of stay.** A linear regression model was trained on 80% of the data, then predicted (on the 20% testing dataset, having not been exposed to this set before) length of stay (A) with an r-squared value of 0.838 with an root mean squared error (RMSE) of 0.108. The actual length of stay was overestimated for longer visits by the linear regression model (B). A random forest model was also trained on 80% of the data, then predicted (on the 20% testing dataset, having not been exposed to this set before) length of stay (C) with an r-squared value of 0.830 and an RMSE of 0.133. The actual length of stay was underestimated for longer visits by the random forest model (D). The relative importance of features the random forest model used were (in descending order): time of day discharged, time of day registered, a presentation in the year 2022, age (younger = longer), a discharge disposition home, a "head and neck" category chief complaints, and a presentation in the year 2021 (E).
(DOCX)

**S1 File. Statistical and modelling packages used.**
(DOCX)

## Author Contributions

**Conceptualization:** Emily Lehan, Peyton Briand, Eileen O'Brien, Aleena Amjad Hafeez, Daniel J. Mulder.

**Data curation:** Peyton Briand, Eileen O'Brien, Aleena Amjad Hafeez, Daniel J. Mulder.

**Formal analysis:** Emily Lehan, Peyton Briand, Eileen O'Brien, Daniel J. Mulder.

**Investigation:** Eileen O'Brien, Aleena Amjad Hafeez, Daniel J. Mulder.

**Methodology:** Emily Lehan, Peyton Briand, Eileen O'Brien, Daniel J. Mulder.

**Project administration:** Peyton Briand, Daniel J. Mulder.

**Resources:** Daniel J. Mulder.

**Supervision:** Daniel J. Mulder.

**Validation:** Daniel J. Mulder.

**Visualization:** Peyton Briand, Daniel J. Mulder.

**Writing – original draft:** Peyton Briand, Daniel J. Mulder.

**Writing – review & editing:** Emily Lehan, Peyton Briand, Eileen O'Brien, Aleena Amjad Hafeez, Daniel J. Mulder.

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
