## [Decision Letter · Decision Letter 0]

24 May 2024

PDIG-D-24-00160

Synergistic patient factors are driving recent increased pediatric urgent care demand

PLOS Digital Health

Dear Dr. Daniel Mulder,

Thank you for submitting your manuscript to PLOS Digital Health. After careful consideration, we feel that it has merit but does not fully meet PLOS Digital Health's publication criteria as it currently stands. Therefore, we invite you to submit a revised version of the manuscript that addresses the points raised during the review process.

Please submit your revised manuscript within 60 days (July 30, 2024). If you will need more time than this to complete your revisions, please reply to this message or contact the journal office at digitalhealth@plos.org. Please include the following items when submitting your revised manuscript:

We look forward to receiving your revised manuscript.

Kind regards,

Cleva Villanueva, M.D., Ph.D.

Guest Editor

PLOS Digital Health

Journal Requirements:

Additional Editor Comments (if provided):

Introduction:

It is recommended to include in the introduction more information regarding specific background (machine learning models)

Methodology.

Study Design: Specify whether the study is observational, retrospective, prospective, experimental, etc.

Population and Sample: Describe the target population and the sample selection process. Include inclusion and exclusion criteria.

Data Collection: Briefly outline the data collection methods (e.g., medical record review, surveys, interviews).

Variables: Identify the primary variables to be analyzed (e.g., demographic characteristics, waiting times, reasons for consultation).

Data Analysis: Indicate the statistical analysis methods to be employed for interpreting the data (e.g., bivariate analysis, multivariate regression).

It is not clear and need to be justified, why the authors did not include visits longer than 1000 min, could longer visits affect the results?

Why did the linear regression model have tendency to overestimate long visits. Was this the reason why visits above 1000 minutes were excluded from the analysis?

Please consider as confounding factors staffing fluctuations and quantity of paperwork, those factors play a role on the length of stay. If data are not available, those factors could be considered a limitation of the study

Results

It is important to include in the explanation of limitations the validity of studying only one center in terms of demography, center capacity in electronic health records

Discussion

Expand the discussion with machine learning models in action

Please be more explicit about how machine learning facilitated this study because it seems that similar analysis can be done without the help of a machine learning software

Please include in the discussion about the importance of machine learning techniques to obtain these results. Discuss how random forest model can be used in the future in medicine for prediction models

Reviewers' comments:

Reviewer's Responses to Questions

**Comments to the Author**

1. Does this manuscript meet PLOS Digital Health’s publication criteria? Is the manuscript technically sound, and do the data support the conclusions? The manuscript must describe methodologically and ethically rigorous research with conclusions that are appropriately drawn based on the data presented.

Reviewer #1: Yes

Reviewer #2: Yes

Reviewer #3: Yes

Reviewer #4: Partly

2. Has the statistical analysis been performed appropriately and rigorously?

Reviewer #1: Yes

Reviewer #2: Yes

Reviewer #3: I don't know

Reviewer #4: Yes

3. Have the authors made all data underlying the findings in their manuscript fully available (please refer to the Data Availability Statement at the start of the manuscript PDF file)?

Reviewer #1: No

Reviewer #2: Yes

Reviewer #3: Yes

Reviewer #4: Yes

4. Is the manuscript presented in an intelligible fashion and written in standard English?

Reviewer #1: Yes

Reviewer #2: Yes

Reviewer #3: Yes

Reviewer #4: Yes

5. Review Comments to the Author

Reviewer #1: The study utilized extensive pediatric urgent care data from a single center to identify factors leading to increased service demand, employing machine learning models to analyze patterns from 2006 to 2022. It identified declining primary care access, circulating infections, and changing complaint types as major drivers of increased visit frequency and duration.

The study has several limitations.

• The findings are based on data from one center, limiting generalizability.

• It does not account for changes in local population demographics, center’s capacity such as staffing or resources and introduction and evolution of Electronic Health Records (EHR).

• The study lacks a clear discussion on the clinical impacts of the observed trends, leaving the practical significance of the findings ambiguous.

• The practical benefits of predicting visits to be in 2022 are unclear.

• It is unclear why the authors filter out the visits longer than 1000 minutes.

Reviewer #2: In this manuscript, the authors highlighted the results of their research aimed to model the factors that have led to the increased pediatric urgent care demand.

This manuscript is well-prepared and presents valuable clinical information for the readers. 

However, I have some critics.

Firstly, I recommend presenting the Methods part with the Results and Discussion

Also, I recommend ammending the Introduction with more information regarding specific background (machine learning models). 

Next, I believe that the Discussion part requires expanding to include more examples of such models in action

Reviewer #3: Synergistic patient factors are driving recent increased pediatric urgent care demand 

Relevance of the study:

The biggest take home message from this study is that it showed the impact of lack of primary care physicians as possible factor for increase of acute care visit. This is particularly pertinent given the substantial deficit of primary care providers, including internists, family medicine practitioners, and pediatricians, observed in the US as well as other advance economies.

Major comments:

• The study does not account for very important factors influencing the length of stay in acute care services, such as evolving documentation practices over time (more paperwork) and staffing fluctuations (bigger patient to provider ratios), particularly notable in the aftermath of the COVID-19 pandemic. These variables should be incorporated into the analysis as potential confounding factors. If data on these aspects are unavailable, authors should acknowledge these as limitations of the study.

• Patient should be more explicit about how machine learning facilitated this study because it seems that similar analysis can be done without the help of a machine learning software. 

Minor comments

• Why did the linear regression model have tendency to overestimate long visits. Was this the reason why visits above 1000 minutes were excluded from the analysis? Was ever considered including patients who had visits of more than 1000 minutes? I am wondering how including those patients would affect the results? 

• Please include in the discussion about the importance of machine learning techniques to obtain these results. Discuss how random forest model can be used in the future in medicine for prediction models.

Reviewer #4: The methods presented in the abstract are not sufficiently clear. While a full dissertation on the methodology is not required, it is recommended to provide a brief methodological description. The methodology of the study is not clearly identified in the methods section.

To enhance the introduction of an article, particularly in terms of methodology, it is crucial to include a succinct description of the approaches and techniques to be used for investigating the hypotheses and achieving the study's objectives. The following methodological elements could be incorporated into the introduction to make it more comprehensive:

Study Design: Specify whether the study is observational, retrospective, prospective, experimental, etc.

Population and Sample: Describe the target population and the sample selection process. Include inclusion and exclusion criteria.

Data Collection: Briefly outline the data collection methods (e.g., medical record review, surveys, interviews).

Variables: Identify the primary variables to be analyzed (e.g., demographic characteristics, waiting times, reasons for consultation).

Data Analysis: Indicate the statistical analysis methods to be employed for interpreting the data (e.g., bivariate analysis, multivariate regression).

For example, it is suggested: "To investigate this, we will employ an observational and retrospective study design, analyzing data collected through NACRS. We will analyze the demographic, clinical, and healthcare system characteristics of patients attending the pediatric urgent care center. Statistical methods will include bivariate and multivariate analyses to identify factors contributing to the increased demand. Our objective is to use high-fidelity NACRS data to model and identify factors contributing to the increased demand for pediatric urgent care at our local urgent care center."

Establishing these considerations is necessary to understand the methodological framework of the article. Although there is good data modeling, it is recommended to include a table that analyzes the factors in the bivariate model with the main variables.

It is advised to present the results concisely using standard analysis tables. This work is relevant, and it is essential to adhere to the journal's presentation format. The article can be published by organizing syntactical aspects and clarifying the presentation of the results. The statistical model appears appropriate, and publication is feasible with the suggested revisions.

6. PLOS authors have the option to publish the peer review history of their article (what does this mean?). If published, this will include your full peer review and any attached files.

**Do you want your identity to be public for this peer review?** For information about this choice, including consent withdrawal, please see our Privacy Policy.

Reviewer #1: No

Reviewer #2: Yes: Dr Vitalii Poberezhets, MD, PhD

Reviewer #3: No

Reviewer #4: Yes: Adan Pacifuentes Orozco

---

## [Editor Report · Decision Letter 1]

4 Jul 2024

Synergistic patient factors are driving recent increased pediatric urgent care demand

PDIG-D-24-00160R1

Dear Dr. Daniel Mulder,

We are pleased to inform you that your manuscript 'Synergistic patient factors are driving recent increased pediatric urgent care demand' has been provisionally accepted for publication in PLOS Digital Health.

Best regards,

Cleva Villanueva, M.D., Ph.D.

Guest Editor

PLOS Digital Health

This guest editor reviewed carefully the new submission and, after checking that the authors follow all the recommendations and made changes, accordingly, go to decision of accepting the manuscript for its publication at PLOS Digital Health.